# Functional Enrichment Analysis of Regulatory Elements

**DOI:** 10.3390/biomedicines10030590

**Published:** 2022-03-03

**Authors:** Adrian Garcia-Moreno, Raul López-Domínguez, Juan Antonio Villatoro-García, Alberto Ramirez-Mena, Ernesto Aparicio-Puerta, Michael Hackenberg, Alberto Pascual-Montano, Pedro Carmona-Saez

**Affiliations:** 1Bioinformatics Unit, Centre for Genomics and Oncological Research, GENYO, Pfizer/University of Granada/Andalusian Regional Government, PTS, 18016 Granada, Spain; adrian.garcia@genyo.es (A.G.-M.); raul.lopez@genyo.es (R.L.-D.); javillatoro@ugr.es (J.A.V.-G.); alberto.ramirez@genyo.es (A.R.-M.); 2Department of Statistics and Operational Research, University of Granada, 18071 Granada, Spain; 3Department of Genetics, University of Granada, 18071 Granada, Spain; eaparicio@go.ugr.es (E.A.-P.); hackenberg@ugr.es (M.H.); 4Bioinformatics Laboratory, Biotechnology Institute, CIBM, Avda. del Conocimiento s/n, 18100 Granada, Spain; 5Data Science & Analytics at IDBS (Danaher Group), 68 Chertsey Road, Woking GU21 5BJ, UK; apascual@idbs.com

**Keywords:** gene set analysis, regulation, web tool, enrichment analysis, functional analysis

## Abstract

Statistical methods for enrichment analysis are important tools to extract biological information from omics experiments. Although these methods have been widely used for the analysis of gene and protein lists, the development of high-throughput technologies for regulatory elements demands dedicated statistical and bioinformatics tools. Here, we present a set of enrichment analysis methods for regulatory elements, including CpG sites, miRNAs, and transcription factors. Statistical significance is determined via a power weighting function for target genes and tested by the Wallenius noncentral hypergeometric distribution model to avoid selection bias. These new methodologies have been applied to the analysis of a set of miRNAs associated with arrhythmia, showing the potential of this tool to extract biological information from a list of regulatory elements. These new methods are available in GeneCodis 4, a web tool able to perform singular and modular enrichment analysis that allows the integration of heterogeneous information.

## 1. Introduction

Functional enrichment analysis, also called gene set analysis (GSA), is a widely used method to analyse high-throughput experimental results. GSA aims to discover biological annotations that are over-represented in a list of genes with respect to a reference background. These annotations are employed to interpret the molecular mechanisms and biological processes that are associated with the experimental condition under study.

There are three main types of GSA: (i) Singular Enrichment Analysis (SEA) that evaluates the statistical significance of individual annotations (i.e., pathways or functional terms) in a list of candidate genes like differentially expressed genes (ii) Gene Set Enrichment Analysis (GSEA) analyses the distribution of the genes associated to a given term in the whole experiment with genes ranked by certain criteria, for example, their fold change and (iii) Modular Enrichment Analysis (MEA) that takes advantage of the inherent relationships among annotations to define sets of correlated terms and evaluates their significance together via SEA or GSEA [1]. Therefore, SEA and MEA evaluates the statistical significance of functional annotations in a set of genes with respect to a reference list, usually, the entire genome, while GSEA analyses the over-representation of genes associated with a given annotation in the top position of the ranked list assuming that not only large changes in gene expression may have significant effects on biological functions but also the contributions of multiple genes with lower differences [2]. Comparative revisions of these methods can be found for example in [3,4]. Additionally, several bioinformatics tools have been developed to perform functional enrichment analysis using these methods [5,6,7,8]. L. Geistlinger et al. provide the most comprehensive benchmarking on 10 major enrichment methods to date [9].

Generally, GSA methods have been developed to analyse genes or proteins but, in the last decade, new omics techniques are generating large datasets for other biological entities such as small RNAs, transcription factors (TFs) or methylation sites [10], that also demand tools for functional characterisation of large lists of such regulatory elements, which has become an active focus of research for the bioinformatics community [2].

As of today, there is a scarce number of software that implements enrichment analysis for regulatory elements, such as TFTenrichr [11] that allows users to perform enrichment analysis of transcription factor target genes, methylGSA [12] for CpGs or miEAA [13] for miRNAs. Nevertheless, most of the GSA tools are developed using gene-based annotations, thus requiring as input the set of target genes associated with the list of regulatory elements, which has an important limitation. Additionally, a bias has been reported in the use of standard enrichment analysis tools in sets of target genes, as these methods have been developed under the assumption that genes are selected uniformly at random from a reference list, which may not always be true [12,14,15,16,17].

In this context, Young et al. studied the gene selection bias from gene expression experiments and its impact in GSA and developed the GOseq method, which was able to account for such bias [15]. Previously published works also report that several of the terms enriched in the analysis of methylation or miRNAs data would also be identified with random data. The gene selection bias depends on the miRNA-target gene interactome or the methylation platform due to a differential distribution of miRNAs or CpGs probes associated with each gene, which consequently favour the terms associated with the genes targeted more often [16,17,18].

In this work, we have developed a singular and modular enrichment analysis tool for regulatory elements, including miRNAs, TFs and CpG sites, besides genes and proteins. The functionality has been implemented in a new release of Genecodis, a widely used application in the field of enrichment analysis [19,20,21]. It applies MEA combining different types of annotations and extracts sets of terms that are jointly associated with a minimum number of genes from the input list using frequent pattern algorithms. Additionally, for the GSA of regulatory elements, it includes the implementation of the Wallenius noncentral hypergeometric distribution, which is defined as an extension of the hypergeometric distribution to the case where the probability of the events differs. In our context, this allows us to overcome the biased selection of target genes derived from a list of TFs, miRNAs and CpGs, whose interactomes and methylation platforms favours the identification of some genes over others.

To the best of our knowledge, this is the first application for functional analysis of miRNAs, TFs, and CpGs along with genes and proteins in a single tool that also integrates the three most common statistical methods for functional analysis of regulatory elements. GeneCodis4 application is available at https://genecodis.genyo.es.

## 2. Materials and Methods

### 2.1. Data Collection

#### 2.1.1. Gene/Protein and Regulatory Elements Data Collection

GeneCodis4 supports the analysis of gene, protein, TFs, CpG sites and miRNAs for 14 different species, main model organisms.

The gene catalogue acknowledged by GeneCodis consists of the commons from NCBI and Ensembl. Ensembl GTFs files are downloaded for each organism and mapped to the NCBI gene archive. In addition, different nomenclatures are available such as gene symbols and Uniprot.

miRNAs identifiers are obtained from miRBase [22]. Their target genes derive from miRTarBase [23] interactome screened only by strong evidence techniques such as reporter assay, western blot and qRT-PCR.

The DNA methylation assays that can be studied in GeneCodis must include CpG sites from the human Infinium MethylationEPIC platform.

Finally, human and mouse TFs-gene interactions were included from DoRothEA [24] with the three highest confidence scoring criteria A, B and C, that report interactions manually curated by experts in specific reviews, supported both in at least two curated databases and by curated and/or ChIP-seq interactions with different levels of additional evidence.

#### 2.1.2. Annotation Data Collection

In GeneCodis4, there are 21 different annotation options found in the database which are grouped into four categories: functional, regulatory, phenotypes and drug databases.

The functional category covers the following databases: BioPlanet [25], Gene Ontology [26,27] in its three subcategories, Biological Process (GO BP), Molecular Function (GO MF) and Cellular Component (GO CC), KEGG Pathways [28], Mouse Genome Informatics database [29], Panther Pathways [30], Reactome [31] and WikiPathways [32]. These databases provide annotations of pathways and biological processes (detailed information can be found via the online help of the application).

The regulatory category contains two aforementioned curated interactomes TF-gene pairs from DoRothEA and miRNA-gene interactions from miRTarBase. Moreover, as a specialised regulatory subcategory, miRNAs-based annotations are added such as the Tool for miRNA Set Analysis (TAM) database [33], the Human miRNA Disease Database (HMDD) [34] and the Mammalian ncRNA-Disease Repository (MNDR) [35]. Unlike MNDR, TAM and HMDD collapse their functional annotations to precursor miRNAs, and in order to facilitate the analysis of different stages of miRNAs, their mature identifier has been included in these two databases as annotated.

Finally, the last categories consist of two types of associations: (1) gene-chemicals from the Comparative Toxicogenomics Database (CTD) [36], the LINCS consortium [37] and PharmGKB [38] databases, and; (2) gene-phenotype from DisGeNET [39], the Human Phenotype Ontology (HPO) [40] and the Online Mendelian Inheritance in Man (OMIM) database [41].

### 2.2. Analysis Workflow and Results Interpretation

#### 2.2.1. Co-Annotations Discovery Algorithm

GeneCodis initially used the Apriori algorithm to find concurrent biological annotations, however, in this last update, these are discovered via the Frequent Pattern algorithm based on tree structures [42]. 

Two Frequent Pattern algorithms are implemented in GeneCodis: FPgrowth and FPmax. The difference between these is the type of co-annotations reported. FPgrowth finds closed co-annotations which are all the combinations of annotations that share a minimum number of input elements. FPmax reports closed co-annotations that are maximal which means they are a superset of closed co-annotations. FPmax is faster and although its results are not redundant they might be difficult to interpret when many terms are co-annotated.

GeneCodis performs SEA by default, the MEA or co-annotation discovery needs to be activated. By default, the algorithm selected is FPgrowth with 10% of the input elements as the minimum number of elements per co-annotation. Notably, it is important to remark that increasing the number of different annotation sources and input elements while decreasing the minimum co-annotated percentage can significantly multiply the number of co-annotations and thus the complexity of the results and its computation time. For this reason, the co-annotation discovery is limited to input lists up to a thousand elements and up to two different sources of annotations per analysis.

#### 2.2.2. Statistics Methods

We have implemented two methods to test the input or the input-targets list against a reference background set: First, the standard hypergeometric distribution test, commonly used in SEA of genes, also called the Fisher Exact test (see Equation (1)). Secondly, the Wallenius noncentral hypergeometric distribution (see Equation (2)), which is preferred to analyse input lists of miRNAs, TFs or CpGs that need to be transformed to their linked genes. Because of the probability of a gene appearing as a target is not uniform, Wallenius uses as weight the odds ratio of the probabilities of the genes being targeted in the category versus the ones outside it [43,44]. Finally, the computed *p*-values are corrected for multiple testing via False Discovery Rate (FDR) from Benjamini and Hochberg [45].

Considering, *x* as the number of elements in the input list that belongs to the tested term, *n* as the total number of elements in the tested term, *N* as the total number of elements in the input list and *M* as the background set, all the elements available for testing, the probability mass function of the hypergeometric distribution follows:(1)$$p\left(x,M,n,N\right)=\frac{\left(\begin{array}{c} n\\ x \end{array}\right)\left(\begin{array}{c} M-n\\ N-x \end{array}\right)}{\left(\begin{array}{c} M\\ N \end{array}\right)}$$

Accordingly, considering *w* as the odds ratio of the tested term, the probability mass function of the Wallenius noncentral hypergeometric distribution is defined as:(2)$$p\left(x,M,n,N,w\right)=\left(\begin{array}{c} n\\ x \end{array}\right)\left(\begin{array}{c} M-n\\ N-x \end{array}\right)\int_{0}^{1}{\left(1-t^{w/D}\right)}^{x}{\left(1-t^{1/D}\right)}^{N-x}dt\;\mathrm{where}\;D=w\left(n-x\right)+\left(\left(M-n\right)-\left(N-x\right)\right)$$

The background set used in both tests can cover two types of scopes: the annotated universe or the whole known measurable elements. The annotated universe fits the distributions considering only the genes regarded by the selected database. If the whole universe is chosen it matches the number of input elements allowed. Nevertheless, the user can introduce a custom input universe that is recommendable, for example, if the user wants to test a list of genes and the upstream analysis is not based on NCBI or ENSEMBL nomenclature.

Furthermore, a relative enrichment score is computed as the ratio between two fractions: (1) the number of genes found in the input for an enriched annotation divided by the size of the input list; (2) all the genes of that annotation divided by all the genes in the database. High scores imply that the annotation is more represented in the input list than in the genome.

#### 2.2.3. Handling the Gene Selection Bias

The gene selection bias appears in common enrichment analyses that are based on the hypergeometric test. This test starts from the assumption that the probability of an element being selected is uniform across the whole experiment. When this condition is not met, several of the terms that seem enriched would also be identified with randomised data [12].

Namely, in RNAseq data, as reported by Young et al. [15], the bias derives from expression analysis in which the expected read count for a transcript is proportional to the gene’s expression level multiplied by its transcript length. Hence, long or highly expressed transcripts are more likely to be detected as differentially expressed compared with their short and/or lowly expressed counterparts. In order to overcome that issue in SEA, they created the R package GOseq that uses a probability weighting function to quantify the probability of a gene being selected that changes as a function of its transcript length and later apply the Wallenius test in each term [15]. GOseq laid the foundation to address biassed SEA, however recent techniques in RNAseq expression analysis, such as RSEM [46], STAR/HTseq-count [47,48], and Salmon [49] reduce the transcript length bias.

The GSA of CpGs, miRNAs and TFs, usually requires the transformation to a list of their associated genes because most of the annotation sources are focused on these. It is during this step where the bias arises for these regulatory elements. Exceptionally, a few databases directly link miRNAs with concrete annotations, such as TAM, HMDD, and MNDR detailed above.

In genome-wide methylation data and arrays, the bias emerges as a result of differences in the numbers of CpG sites associated with different classes of genes and gene promoters. It is due to differences in the number of methylation probes that are associated with each gene [16]. Then, for this type of data, separate methods adjusting for the number of CpGs instead of gene length are necessary [12].

For miRNAs and TFs the bias occurs depending on the interactome with their target genes. The miRNAs interactome is mainly studied in oncology research, causing cancer and cell cycle terms to be often enriched when performing the GSA over their target genes [17]. Therefore, in the GSA analysis of miRNAs and TFs the probability of a gene being selected depends on the number of TFs and miRNAs that regulate it.

To address this limitation, the approach proposed by GOseq is implemented in GeneCodis4. In which, first, a bias score of each gene is calculated through a probability weighting function [15]. This function derives from a logistic GAM model that fits a cubic spline with a monotonicity constraint to a binary data series where a value of 1 refers to an input gene or TF-, miRNA-, CpG-target gene and 0 if none. The fitting is against different variables depending on the type of input. If genes are introduced, GeneCodis4 uses their length, in the case of CpGs the bias is given by the number of CpGs associated with each gene, likewise, for TFs and miRNAs is the number of TFs or miRNAs per target gene [15,44]. Then the Wallenius test uses the odds ratio for each term (*w*) calculated as the average gene bias score of the annotated elements in the category divided by the average of those that are not.

Lastly, for miRNAs when using the standard hypergeometric test, GeneCodis4 implements the third approach of Godard and van Eyll [17] where gene-based databases are transformed into lists of miRNAs that target at least one of these genes. This strategy ensures that a miRNA is only represented once in a category whatever the number of its target genes is. Additionally, they also show that many pathways share a significant number of miRNAs often leading to their co-identification thus recommending their aggregation before the GSA for which the MEA of GeneCodis is prepared.

To sum up this section, GeneCodis4 offers three approaches to handle the bias. The Wallenius test approach is available for all types of input, meanwhile the other two, the use of direct annotations and the database transformation are only available for miRNAs.

#### 2.2.4. Results Visualisations

The results report provides an interactive table to explore the top 100 enriched terms and the complete result table can be downloaded. Two interactive, customizable and downloadable visualisations are generated, a network where annotations are linked to their genes or miRNAs and a bars chart. The network has a profuse force layout which causes annotations to be clustered as they share more genes. The size of the annotation node or bar is proportional to −log10 (*p*-value adj.).

### 2.3. Web Application Implementation

The back-end is developed with Python 3.8 and the Flask microframework to create an application programming interface (API). Via the python library Gunicorn, a WSGI HTTP Server for UNIX, the API is deployed. Finally, the GeneCodis4 web page is sent to the client-side with an NGINX server while also acting as a front-end reverse proxy of the application. The database is built with PostgreSQL 12 and accessed by the psycopg2 python module.

The Frequent Pattern algorithms are implemented from the library MLxtend [50]. The probability weighting function is obtained with pyGAM [51]. The stats methods are built-in functions of Scipy [52] and the *p*-value correction is found in the statsmodels package.

The GeneCodis4 front-end is built using plain HTML, JavaScript and CSS. HTML pages are rendered with EJS, a JavaScript templating language. The interactive table is displayed by the DataTables plug-in. Visualisations depend on two JavaScript libraries, D3.js and jQuery. The CSS mainly derives from the Bulma framework.

GeneCodis4 is deployed in Ubuntu Server 18, in a computer with 252 GB of RAM memory and a microprocessor Intel(R) Xeon(R) Silver 4214R CPU @ 2.40GHz.

## 3. Results

### 3.1. Analysis of Arrhythmia Related miRNAs

The proposed methods are compared by analysing a set of arrhythmia-related miRNAs. The selected miRNAs (miR-1, miR-133, miR-328, miR-212, miR-208a) are extracted from Table 1 in a review that studies miRNAs linked to cardiac excitability and other processes pertinent to arrhythmia [53].

The selected list of miRNAs needs to be transformed into proper miRNAs identifiers. Following the latest miRBase annotation, miR-1 family contains precursors mir-1-1 and mir-1-2, likewise, the miR-133 family is divided into mir-133a-1, mir-133a-2 and mir-133b. These precursors are added to the input list along with the precursor mir-328, mir-212 and mir-208a. Last, they need the human prefix (hsa-) to follow the miRBase identifiers convention and to be mapped in the GeneCodis database.

Finally, since gene-based annotations are selected and only mature miRNAs are associated with targets, the list of miRNAs precursors obtained before needs to include their mature form. This is done by using the GeneCodis miRNAs converter tool to incorporate all the mature or precursor forms missing. This converter is visible below the input box when selecting miRNAs input type. The resulting list of miRNAs is the provided example in the website when selecting human and miRNAs as input to facilitate the recreation of this use case.

The analysis consists of a Singular GSA with the annotated background set scope using one database of each of the four annotation categories available in GeneCodis4 and different approaches:
(a)Wallenius test of the miRNAs target genes(b)Hypergeometric standard test of the transformed miRNAs-based annotations(c)Hypergeometric standard test of the direct miRNAs-based annotations(d)Hypergeometric standard test of miRNAs target genes

The target genes of the miRNAs are obtained from the “Annotated genes” in the quality control report from the results of strategy (a). In order to estimate the biological significance for each approach, the results should recover terms highlighted in the conclusions of the miRNAs set source article. These terms should reflect properties of cardiac excitability including conduction, repolarization, automaticity, ion Ca2+ and other ion channel regulation, spatial heterogeneity, and apoptosis and fibrosis. In Table 1 their occurrences within the top 20 are counted. The complete results tables are provided in *excel* file Appendix A.

In pdf file Appendix A shows the overlap of the heart related terms counted in Table 1 for each comparable annotation and strategy.

Gene Ontology Biological Process results are more specific in using the transformed databases with terms directly related to the regulation ion channels and membrane repolarization i.e., “positive regulation of potassium ion transmembrane transport” and “membrane repolarization during ventricular cardiac muscle cell action potential”, unlike the strategies based on target-genes, Wallenius and Hypergeometric test, that report broader terms as most significant i.e., “negative regulation of apoptotic process”, “heart development”. Besides, in these last approaches most of the terms include regulation of cell cycle, transcription and gene expression. Figure 1 shows a network plot of the top 20 most significant terms (Wallenius *p*-value < 0.05) of the analyses with GO Biological Process.

The results of strategy based on miRNAs databases are specific to the MNDR database, since the others are gene-based. Its top results contain 15 heart diseases within the top 20, see Figure 2.

The PharmGKB results with the three available strategies contain antiarrhythmics, such as “quinidine” and “amiodarone” plus calcium channel blockers “nitrendipine” and “felodipine”, although in the hypergeometric test of the target genes no term is found significant. In the target gene approaches, “warfarin”, an anticoagulant drug, is found in the top 20 and not in in the database transformed annotations to miRNAs. In this strategy, however, the top significant drug is the antiarrhythmics “dofetilide” next to the general term “calcium channel blockers” and its derived drug “mibefradil”, “lidoflazine” is HERG K+ channels blocker, and “doxazosin” is used in antihypertensives treatments. Among others, enriched drugs “prenylamine”, “grepafloxacin”, “levomethadyl acetate” or “moxifloxacin” induce heart-related disorders, torsades de pointes and Long QT Syndrome. These results are displayed in the miRNAs-annotation network plot in Figure 3.

Regarding the results of HPO, surprisingly, the standard hypergeometric test of the target genes counts more heart-related disorders than the Wallenius appraoch. “Ventricular septal defect” is unique of Wallenius while it shares with the standard Hypergeometric “Syncope”, “Prolonged QTc interval”, “Abnormal T-wave” and “Ventricular fibrillation”. The other four phenotypes in the Hypergeomtric are: “Shortened QT interval”, “Torsade de pointes”, “Sinus bradycardia” and “Abnormal cardiac exercise stress test”. These terms or similar terms are also found in the transformed HPO database where 17 out of 20 are enriched with cardiac pathologies.

Network and bar chart figures for the remaining results are included in Appendix A. Notwithstanding, the reader can easily recreate the analysis in GeneCodis in order to explore the results more in detail selecting the human miRNas example.

Although using Wallenius statistics reduces the bias in the enrichment analysis as has been reported previously, we still observe that annotations associated with a high number of target genes may ubiquitously appear as significant terms. Therefore, more specific annotations of miRNAs will allow us to have more specific results. In this context, GeneCodis offers two miRNAs-based annotations focused on diseases that can provide very interesting information in the analysis of disease biomarkers. If researchers are interested in the analysis of information about drugs, pathways and other biological processes, the analysis should be done with the other two proposed approaches.

From these use case results, it is expected that the reader will become aware of the importance of using the proper approach when performing GSA of regulatory elements.

### 3.2. Integration in a Complete miRNAs-Seq Analysis Suite

GeneCodis4 is included within SRNAToolBox-sRNAbench [54]. By means of this integration, a complete analysis pipeline can be carried out. By using SRNAToolBox, the user could perform sRNA expression profiling from next-generation sequencing data in order to obtain a set of candidate miRNAs that can then be functionally characterised in GeneCodis4 in a straightforward manner.

## 4. Discussion

Enrichment analysis is a common approach to extract biological knowledge from omics experimental results. These methods have been widely used for analysing gene or protein lists, for which GeneCodis is a well-established tool. Besides, GeneCodis4 offers new possibilities to analyse other biological entities such as miRNAs, TFs and CpGs. The functional characterisation of these regulatory elements has been normally inferred via their target genes but this approach has proven a gene selection bias that limits the interpretation of the results. We have implemented three different approaches in GeneCodis4 to overcome that issue, which are the Wallenius statistics [15] and the use of miRNAs specific annotations, using direct annotation databases [33] or a transformation of gene annotation databases to miRNAs based annotations [17].

These have been applied in the use case analysis of arrhythmias miRNAs in order to understand the advantages and limitations of each one. It must be highlighted the importance of selecting the proper GSA statistics and the underlying annotations for each experimental setting to recover biological meaning, for which GeneCodis4 offers settled approaches in a single tool.

Most of the enrichment analysis tools implement only SEA, a single bias handling method, and normally only accept a single type of biological entity, furthermore, they normally require bioinformatics skills. Hence, GeneCodis4 is presented as a unique web tool for the SEA and MEA for the functional characterisation of genes, proteins, miRNAs, TFs and CpGs.

In addition, GeneCodis is one of the few applications that allows modular enrichment analysis, which can provide useful information in the analysis of regulatory elements with other sources of information such as biological processes or pathways. In the analysis of regulatory networks of TFs and miRNAs, it is important to reflect their expression levels for which the upload of lists of up-regulated and down-regulated miRNAs or TFs in a separated analysis allows us to discover functions that are enriched in association with up-regulation or down-regulation. Similarly, the user can divide TFs under study into the analysis of activators and repressors, to understand which pathways or functions are being enhanced or diminished.

GeneCodis4 was released in February 2020 and, since then, it has received an average of 300 unique users per month, continuing with the previous trend that has positioned GeneCodis as a reference tool in the field of GSA.

These new functionalities place GeneCodis4 as an excellent web tool that requires no bioinformatics skills to integrate results from different omics and knowledge perspectives providing the state-of-the-art in the GSA of regulatory elements.

## Figures and Tables

**Figure 1 biomedicines-10-00590-f001:**
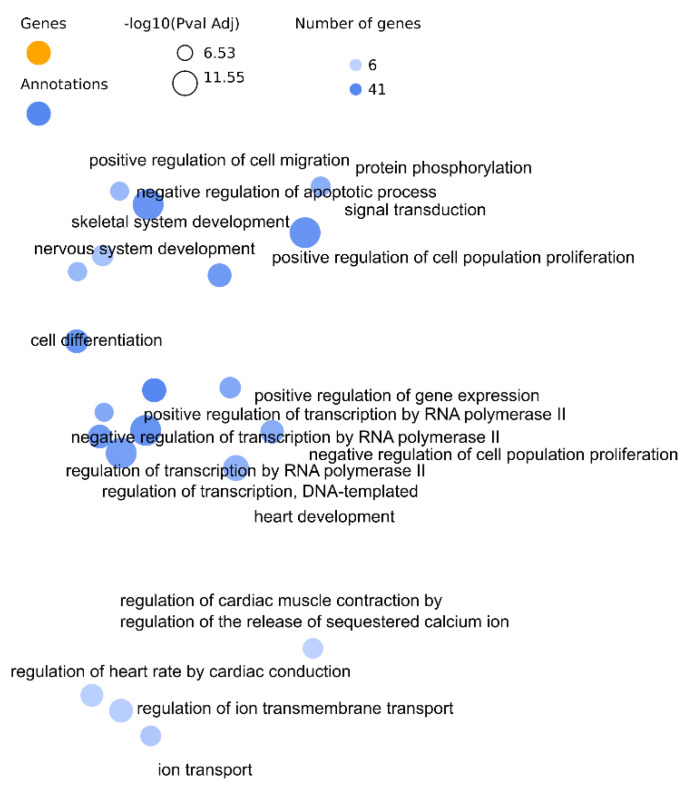
Network plot with genes hidden in the GO BP results of the use case with Wallenius strategy. Three clusters can be observed from top to button, the first related to cell cycle regulation, the second to gene regulation and the last one contains cardiac excitability biological processes.

**Figure 2 biomedicines-10-00590-f002:**
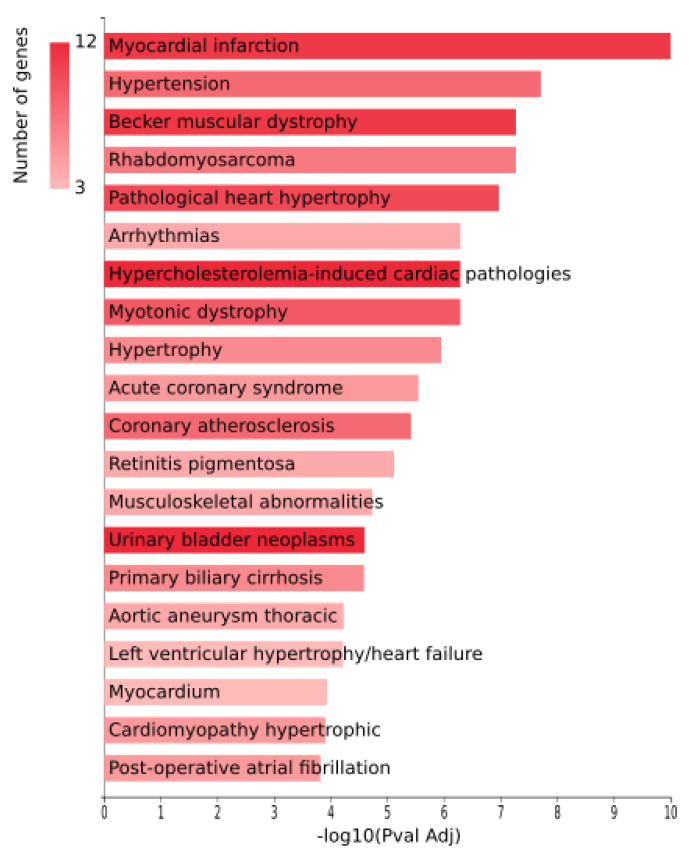
GeneCodis4 bars chart plot of the MNDR database in the direct annotation of miRNAs strategy. 15 out of 20 are heart-related disorders.

**Figure 3 biomedicines-10-00590-f003:**
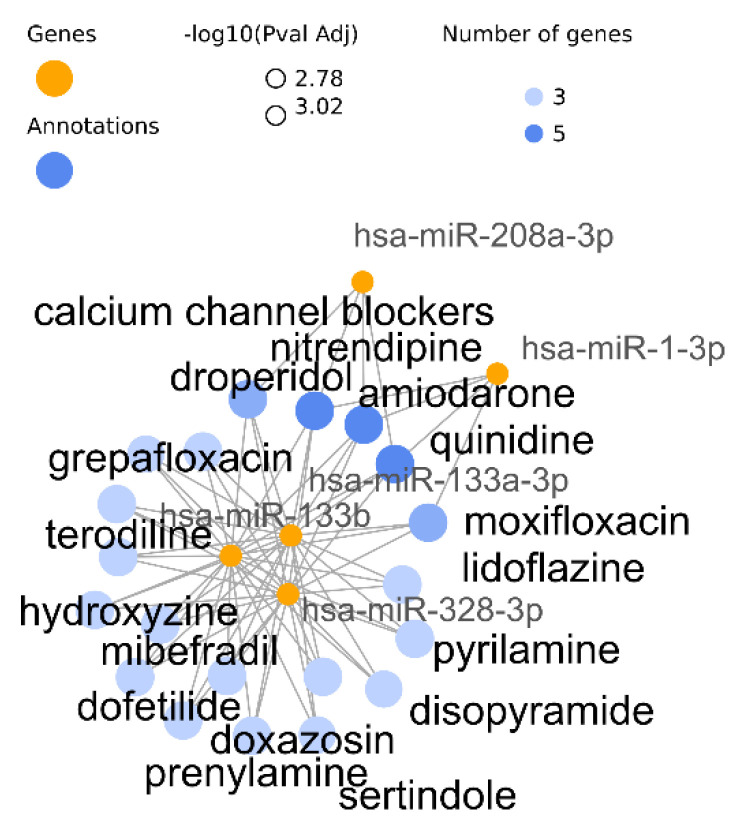
Show the top 20 terms enriched with the transformed database strategy. Only five different miRNAs cause the enrichment of these terms, being four antiarrhythmics associated with all of them.

**Table 1 biomedicines-10-00590-t001:** Count of cardiac excitability associated terms in each type of analysis and annotation database in the top 20.

Approaches	GO BP	PharmGKB	HPO	MNDR
Wallenius target-genes	7	9	5	-
Transformed DBs	20	19	17	-
miRNAs-based DBs	-	-	-	15
Hypergeometric target-genes	4	6	8	-

## Data Availability

Data sharing not applicable.

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
