# Peer review of "Functional Enrichment Analysis of Regulatory Elements"

_biomedicines, 2022, doi:10.3390/biomedicines10030590_

Round 1
Reviewer 1 Report
1- i.e (i.e.,) italic also et al, should be (et al.,) also italic
2- "There are three main types of GSA: i) Singular Enrichment Analysis (SEA) that evaluates the statistical significance of individual annotations (i.e pathways or functional terms) in a list of candidate genes ii) Gene Set Enrichment Analysis (GSEA) analyzes the distribution of genes associated to a given term in the whole ranked list of genes and iii) Modular Enrichment Analysis (MEA) that takes advantage of the inherent relationships among annotations to define sets of correlated terms and evaluates their significance together via SEA or GSEA [1]". Please explain the relationship between these three aspects, what is the level of redundancy between them, and can they be applied to all gene lists? Why would a reader using this method use all three aspects and what are the advantages?
3- "development biology", developmental biology.
4- There is a shift between the introduction of the gene ontology method and then a paragraph about what seems to be a web of terms such as gene regulatory elements, TF, methylation site, and miRNA, making it difficult to ascertain the exact focus of the second paragraph. Could be sharpened.
The third paragraph investigates the bias of gene lists produced by RNA-seq and miRNA approaches. In my opinion, the introduction could be written more precisely as it is convoluted at present.
5- "It applies MEA combining different types of annotations and extracts sets of terms that are jointly associated with a minimum number of genes from the input list using frequent pattern algorithms." How does this method improve the background model against which the enrichment scores are calculated?
6- "It includes the implementation of Wallenius noncentral hypergeometric distribution to overcome the bias selection of target genes derived from a list of TFs and miRNAs whose interactomes and methylation platforms favours the identification of some genes over others". Please define the Wallenius model.
7- "Functional category covers the following databases: BioPlanet [23], Gene Ontology [24,25] in its three subcategories, Biological Process (GO BP), Molecular Function (GO MF) and Cellular Component (GO CC), KEGG Pathways [26], Mouse Genome Informatics database [27], Panther Pathways [28], Reactome [29] and WikiPathways [30]." It is not clear what the authors are doing with these databases, please explain in more depth. If these were used for data annotation, how were they used? This applies to all the tools in 2.1.1.
8- "Additionally, motivated by the necessity to assist global research on the SARS-CoV-2, the GO specific subset is included covering functions of proteins used by the virus to enter a human cell." Off-topic.
9- "It is important to note that increasing the number of different annotation sources and decreasing the minimum number of genes can significantly multiply the number of co-annotations and thus the computation time." What is the advantage of this aspect?
10- "We have implemented two methods to test the input list against a reference background set: the standard hypergeometric and the Wallenius noncentral hypergeometric distributions. The first one is commonly used in GSA while the second approach is preferred to analyse input lists of miRNAs, TFs or CpGs that need to be transformed to their linked genes [41,42]." These terms were used before and can be elaborated on (e.g., background about standard hypergeometric model etc).
11- The p in p-value should be italicised
12- "Namely, in RNAseq data, as reported by Young et. al. [14] the bias derives from differential expression analysis in which the expected read count for a transcript is proportional to the gene's expression level multiplied by its transcript length. Hence, long or highly expressed transcripts are more likely to be detected as differentially expressed compared with their short and/or lowly expressed counterparts". Recent methods for analysing RNA-seq have reduced transcript length bias such as RNA-Star/ HTseq-count and Salmon.
13- "Exceptionally, there are a few databases that directly link miRNAs 191 with concrete annotations". Such as?
14- "In genome-wide methylation data and arrays, the bias emerges as a result of differences in the numbers of CpG sites associated with different classes of genes and gene promoters". Please explain how.
15- "The selected list of mature miRNAs needs to be transformed to proper miRNAs identifiers because miR-1 and miR-133 refer to different families that share the same root sequence. Thus, following the latest miRBase annotation miR-1 family contains mir-1-1 and 253 mir-1-2, likewise, miR-133 is divided into mir-133a-1, mir-133a-2, and mir-133b". Have any miRNAs been added as a result of this conversion?
16- Table 1 requires a clearer explanation in the main text (e.g., why certain categories could not perform the test) and how modules a-d performed differently, and what the overlap was.
17- Is figure 1 results linked to table 1? It is not clear from the way this is written.
18- The experiments in Figures 1-3 should be shown for all modules if possible to allow for comparisons.
19- The discussion is short and could be substantiated to clarify the message of this paper. At present, there isn't a clear link between some of the topics discussed in the introduction with the methods/ results (e.g., aspects about methylation, TFs, regulatory modules seem to get lost in later parts of the paper). This paper could have been written more concisely avoiding off-topic aspects/ tangents.
Reviewer 2 Report
In this article, the authors have established a set of enrichment analysis methods for regulatory elements, including CpG sites, microRNAs and transcription factors. Additional suggestions for improvements are as follows:
- The introduction section can be improved to highlight the novelty of the article.
- The various limitations of new methodologies that have been applied for the analysis of a set of miRNAs associated with arrhythmia should be analyzed.
- The authors should provide their own justification and relevance of the study. This will help the readers to understand the importance of the paper.
- The discussion section should be further elaboarted and improved.
- All sections of the manuscript should be checked for typographical errors.
Reviewer 3 Report
The authors provide an analysis tool that allows to obtain a lot of information by simultaneously querying of numerous databases. The statistical analysis is robust and could produce valid results even when there are no initial assumptions by data filtering of integrated different sources, annotations and concurrent patterns information.
The interface created is intuitive, however the recovery of the initial information is difficult.
For example, the desciption table shows the number of genes found but there is no direct link on the graph or a summary table that can allow direct selection, from the input list, of genes on which the operator wants to focus attention.
About miRNAs and TFs, there are no information relating to the increase or decrease in expression. This information is necessary for understanding of the network correlations. It could be implemented by coloring the nodes differently (up and down regulation).
Reviewer 4 Report
This study presents a new way of gene set analysis for regulatory elements such as CpG sites and microRNA. It is an interesting, well written manuscript with clearly presented methods. I would only suggest the authors to expand the discussion section and indicate the possible use of this research in clinical practice.
Round 2
Reviewer 1 Report
The authors have addressed my comments.